# Choroid Plexus: The Orchestrator of Long-Range Signalling Within the CNS

**DOI:** 10.3390/ijms21134760

**Published:** 2020-07-04

**Authors:** Karol Kaiser, Vitezslav Bryja

**Affiliations:** Department of Experimental Biology, Faculty of Science, Masaryk University, 625 00 Brno, Czech Republic

**Keywords:** cerebrospinal fluid, choroid plexus, secretion

## Abstract

Cerebrospinal fluid (CSF) is the liquid that fills the brain ventricles. CSF represents not only a mechanical brain protection but also a rich source of signalling factors modulating diverse processes during brain development and adulthood. The choroid plexus (CP) is a major source of CSF and as such it has recently emerged as an important mediator of extracellular signalling within the brain. Growing interest in the CP revealed its capacity to release a broad variety of bioactive molecules that, via CSF, regulate processes across the whole central nervous system (CNS). Moreover, CP has been also recognized as a sensor, responding to altered composition of CSF associated with changes in the patterns of CNS activity. In this review, we summarize the recent advances in our understanding of the CP as a signalling centre that mediates long-range communication in the CNS. By providing a detailed account of the CP secretory repertoire, we describe how the CP contributes to the regulation of the extracellular environment—in the context of both the embryonal as well as the adult CNS. We highlight the role of the CP as an important regulator of CNS function that acts via CSF-mediated signalling. Further studies of CP–CSF signalling hold the potential to provide key insights into the biology of the CNS, with implications for better understanding and treatment of neuropathological conditions.

## 1. Introduction

A singular feature of the CNS, crucial to its function following neural tube closure and homeostasis throughout adulthood, is the presence ofCSF, which represents the key signalling interface between various distant regions of the CNS. CSF occupies an intricate and interconnected network of ventricles and cavities spanning the whole CNS that collectively give rise to the brain ventricular system. This complex component of CNS architecture consists of four brain ventricles—the paired lateral ventricles, 3rd ventricle, and 4th ventricle, that are connected to the central canal of the spinal cord and the subarachnoid space [1]. Preservation of a tubular system filled with CSF as a hallmark of the CNS is characteristic for the whole phylum Chordata [2], including mammals and its evolutionary significance is further highlighted by a similar arrangement consisting of a CNS bathed in a fluid observed even in non-vertebrate organisms such as *Drosophila* [3].

For most of history, CSF has been assumed to primarily act as a fluid cushion providing mechanistic protection to the brain, an osmotic buffer system, and a route for clearance of metabolic waste and toxic compounds from the brain. This rather narrow view has been recently challenged by the growing evidence pointing to the expanded role of CSF as a conduit for delivery of instructive cues involved in the regulation of multiple aspects of CNS embryogenesis, adult neurogenesis, and modulation of adult brain function [4]. These findings helped to shed new light on the previously-underappreciated capacity of CSF to harbour various bioactive compounds and promote long-range signalling in distinct regions of the CNS [5,6], putting into the spotlight distinct parts of the CNS involved in the production of CSF, including the CP [7].

The CP is a secretory tissue located within each of the brain ventricles that is present in all vertebrates [8]. Considering the strategic position of the CP in the CNS and the emerging understanding of its role in the active release of various growth factors and other biologically-active substances into the CSF, the CP has been attracting growing interest as a vital hub orchestrating various aspects of intercellular communication via CSF in both the embryonic and the adult CNS [4,9]. Further solidifying this concept of the CP acting as a signalling centre, recent findings point to its function as a key entry point for signalling complexes from the blood circulation [10].

In this review, we aim to provide an overview of recent advances regarding the composition of CSF from the standpoint of CP secretome and its multifaceted impact on the regulation of various aspects of CNS embryogenesis and maintenance in adulthood.

## 2. CSF—An Intrinsic Component of CNS Environment

Specific aspects of neural tube development in vertebrates permit early and complete separation of CSF from the surrounding environment, thus allowing for precise regulation of its content early on during embryogenesis [11]. The importance of tight control over CSF composition is evidenced by the rapid acquisition of barrier-like properties in early stages of development by all CNS interfaces in direct contact with CSF, which are thus able to specifically shape and fine-tune CSF content and signalling properties [12,13]. Dynamic changes in the embryonic CSF (eCSF) composition and properties mirror dramatic morphological and functional changes occurring in parallel during CNS development. Upon neural tube closure in mammals, the captured amniotic fluid becomes the nascent CSF. In the ensuing period, preceding formation of the CP, it has been shown that CSF composition correlates to large extent with proteomic changes observed in the developing neuroepithelium [14,15], which displays barrier-like properties [16], enabling regulated release of growth factors and particles from neuroepithelium into the CSF [17,18]. Interestingly, the ability of neuroepithelium to tightly control CSF composition in early brain development exhibits interspecies differences, indicating existence of distinct requirements for CSF regulation depending on the complexity of the developing CNS [19,20]. In later stages of the embryonic development, the importance of the neuroepithelium as the key site for active regulation of CSF content decreases as it gradually loses its barrier properties [19]. In late embryogenesis, the CP becomes the major site for the production of CSF and the key player in the regulation of CSF proteomic content and capacity to modulate various biological processes in the CNS [21,22]. Even after CP emergence as the predominant source of CSF, the contribution of other CNS domains, such as ependymal cells lining the ventricle and spinal canal, both during embryogenesis and in adulthood, to the production of CSF cannot be discounted [23,24].

Apart from other functions related to the mechanical protection and metabolic turnover, CSF constitutes a signalling environment indispensable for proper growth and functional maturation of the CNS. eCSF has been demonstrated to be crucial for proper execution of genetic programs underlying embryogenesis of various CNS regions as well as control of adult neurogenesis [25,26]. A growing list of signalling molecules and growth factors identified in eCSF [5,27], provides compelling evidence explaining the disruption of CNS embryonic development observed as a consequence of CSF removal [28]. Furthermore, obstructing CSF flow has been shown to impede distribution of supramolecular complexes such as lipid particles, thus highlighting additional mechanism through which CSF-dependent distribution of various biologically-active compounds and complexes affects embryonic growth [29]. CSF proteome undergoes age-dependent changes [30], likely reflecting distinct requirements for the signalling molecules in the regulation of proper function of the CNS during embryogenesis and adulthood [6,31]. For example, changes in the levels of CSF-borne factor IGF-II, correlate with the age-matched ability of CSF to promote neural proliferation and survival of neural progenitors [5].

In addition to its role as a vehicle for distribution of bioactive molecules, hydrodynamic forces associated with CSF flow establish instructive cues that are able to activate ion channels expressed by adult neural stem cells (NSCs) in direct contact with CSF, which can act as mechanosensory receptors, regulating adult neurogenesis [32]. Interestingly, CSF dynamics change in response to different physiological states such as sleep, implying a possible role of CSF in the coordination of the biological activity across the CNS in response to the altered physiology [33].

The ability of the4 CSF to reach and affect processes in various regions of the CNS is not limited to the cell populations in direct contact with the ventricular space. Instead of simply being drained into the blood stream via arachnoid villi granulations [34], it has been shown that a substantial amount of CSF enters brain parenchyma along paravascular spaces as a part of a recently-described “glymphatic” system [35]. This allows CSF-borne substances to spread to large number of brain regions not connected to the ventricular system [36]. Moreover, recent findings have shown the capacity of CSF-derived proteins to regulate proliferation in the subgranular zone (SGZ), one of the two main neurogenic niches in the adult brain, highlighting the potential of CSF to directly control biological processes in brain regions considered to lack direct access to the CSF [37,38].

Given the function of CSF as an essential route for the long-range trafficking of factors across the CNS, CSF has been also explored as a potential source for biomarkers providing information allowing for early detection and diagnosis of distinct pathological conditions such as neurodegenerative diseases or various types of brain cancer [39,40].

Taken together, CSF constitutes crucial element of the CNS extracellular microenvironment with increasingly appreciated roles that extend beyond being a simple mechanic buffer or drainage system for the brain metabolism. A flurry of recent discoveries revealed CSF as a signalling nexus distributing and integrating signals, consisting of a wide array of bioactive compounds, within the whole CNS.

## 3. The Choroid Plexus—Key Regulator of CSF Production

The choroid plexus (CP) is a secretory tissue protruding into the lumen of all brain ventricles, namely the lateral ventricle CP (LV CP), the 3rd ventricle CP, and 4th ventricle CP (4V CP), in the form of a sheet of epithelial cells that are in direct contact with the CSF and encapsulate richly-vascularized stroma [22]. Unlike other developing processes, CP development progresses in a posterior to anterior manner with 4V CP being first to develop, followed by LV CP with 3V CP being last to emerge [7]. The CP arises from progenitor cells, specified early in the development [41], that are distributed along the dorsal midline and rhombic lip in the case of 4V CP [42,43]. CP epithelium (CPe), which originates in the neuroectoderm [44], forms a monolayer of polarized cuboidal cells with high expression of various transport proteins indicating robust secretory capacity [45,46]. Signalling from the CPe is instrumental for the induction of differentiation of the underlying CP mesenchyme and their mutual interaction is further required for proper choroid plexus morphogenesis [47,48]. Moreover, the CP is populated by additional cell types including immune cells and neurons, indicative of CP functional versatility [49,50].

As such, the CP represents a complex tissue that fulfils distinct roles essential to the CNS function. Several lines of evidence clearly established the CP as the major site for CSF production [21], despite some controversy still remaining regarding the extent of its contribution [51]. Importantly, ablation of various channel and transporter proteins located at the apical side of the CPe resulted in severe decrease in the CSF production providing compelling evidence for role of the CP in this process. Furthermore, the CP has been implicated in the CNS homeostasis via maintenance of CSF pH balance and ion osmoregulation [46]. Along the same lines, the CP actively contributes to the removal of harmful compounds originating from the blood stream or generated by brain metabolism [52,53].

However, the key functional feature of the CP, conferred by the presence of junction proteins in the epithelium [12,45], is the ability of the CPe to act as a selectively-permeable interface, preventing free passage of compounds between CSF and the blood, thus establishing the blood–CSF barrier (BCSFB) [4]. This functional aspect of CP biology is essential. Fenestrated capillaries in the CP stroma and substantial local blood flow rate collectively create a highly-permeable environment enabling fast and unhindered spread of substances from blood to the CP stroma [54]. Significant protein secretion capacity displayed by CPe in tandem with selective transport of compounds from the blood stream might explain the differences of proteomic profiles between CSF and blood [55,56]. Due to its convoluted morphology and presence of microvilli on the apical surface, CPe surface area corresponds up to 50% of the overall luminal area of brain capillaries establishing the blood–brain barrier (BBB) [57,58]. Upon their maturation, CP epithelial cells manifest increased mitochondrial density, thought to provide energy supply for the considerable metabolic demands linked to the secretory activity of the CPe [58,59].

Despite shared morphology and function, embryonic CPs preserve their specific domain identities. They reflect position of the CP along the midline axis and underlie distinct transcription signatures and heterogeneous proteomic profiles observed between different embryonic CPs [49,60]. Intriguingly, secretome differences revealed across embryonic CPs are suggestive of spatially specific gradients of signalling molecules that lead to the localized activation of downstream signalling pathways within the brain. This site-specific effect of various CP-derived regulators further combines with the compartmentalization of CSF flow within the ventricular system caused by ciliary beating or bodily movements [61]. Indeed, SHH and Wnt-5a ligands, both selectively enriched in the embryonic 4V CP, have been recently linked to the modulation of proliferation and tissue patterning in the adjacent cerebellum [62,63]. Importantly, regionalized proteomic profiles may also underlie morphological differences between individual CPs as they have been implicated in different aspects of tissue morphogenesis such as the maintenance of specific progenitor domain associated with the embryonic 4V CP-derived SHH or control of epithelial branching via action of Wnt-5a [64,65]. In addition, observed molecular heterogeneity is associated not only with embryonic epithelial cells but was identified also in other cell populations of developing CPs including fibroblasts, possibly adding another layer to the complexity and specificity of the CP secretory repertoire [49]. Of interest, domain-specific differences in molecular make-up of CPs are also preserved in adulthood. For example, *Sod3* gene expression, encoding a metabolic enzyme, is limited to the adult 4V CP, whereas the expression pattern for protein kinase encoded by the *Penk* gene, is completely reversed [60]. The age-dependent shift in the expression of various genes underlying CP detoxification or CSF production capacity has also been observed [45,52], revealing the dynamic nature of the CP secretory profile over time. It, however, seems that the importance of CSF-borne bioactive molecules released by CSF gradually decreases with age. This view is supported by the general decline in the CPe gene expression in adulthood and the gradual reduction in the CSF vs. brain tissue ratio [12,60]. There have also been recent findings showing suppressed ability of CSF to promote neurogenesis correlated with age-dependent changes in the CP secretome [5,6]. Interestingly, secreted protein Klotho associated with significant anti-aging properties is highly expressed by the CP during early development and adulthood and its CSF levels exhibit gradual decrease during aging [66,67]. Overall, it is possible that this altered pattern of CP secretory activity may reflect more general changes in CNS biology at different stages of life.

Another emerging aspect of CP function is the intrinsic ability to sense and respond to changes in the CSF as well as broader physiological changes. It has been recently shown that the CP expresses genes encoding components of circadian clock machinery, such as *Bma1*, *Per1*, and *Per2*, which allow the CP to influence activity of the key hypothalamic centre involved in the sleep/wake rhythmicity via secreted signals carried by CSF [68,69]. Remarkably, this mode of circadian clock regulation displays sex differences mediated by estrogen signalling [70], which is in line with the previous findings showing sex-based variability in the CP gene expression profile and proteomic signature [71]. It is noteworthy that it has been recently reported that the CP might be, at least partially, involved in the contextual fear-learning as it exhibits, in some instances, stronger response to stressful stimuli at the levels of gene transcription as compared to the hippocampus. Altered expression of multiple genes encoding secreted molecules such as the putative hormone augurin represent an example [72,73]. Recently, the CP has been also implicated as an entry site for various hormones produced in response to changed physiological state that are present in the blood, thus directly affecting their availability in the brain. For example, expression of the receptor for the peptide hormone leptin in the CP, which is involved in the regulation of the fat balance in the body, has been shown to be the limiting step, determining the transport rate of leptin from the blood stream into the CSF [74].

Due to the expression of specific receptors, the CP has been also shown to respond to the presence of neurotransmitters present in CSF such as serotonin or nicotine, which are able to elicit robust changes in the CP metabolism and transcriptome [75,76]. In addition, a recent pioneering study, leveraging a new technique for real-time monitoring of CP activity allowed characterization of a novel mode of apocrine secretion from the CPe in response to stimulation via a serotonin receptor agonist [77].

CSF plays an important role as the key modulator of neuroinflammation. CSF contains distinct pools of activated immune cells, which can be enriched in various neurodegenerative diseases such as Alzheimer’s disease (AD) [78,79]. Moreover, CSF displays a complex profile of cytokines and chemokines, which changes dynamically in different neuropathological conditions [80,81]. Interestingly, CSF-mediated regulation of neuroinflammatory response is shaped by the glymphatic system that serves as an important route for drainage and active clearance of immunomodulators and immune cells present in the CSF [82]. Given the profound changes of the adult CP transcriptome in response to inflammatory stimuli, the CP has recently emerged as an active sensor participating in immunosurveillance within the brain that is capable of dramatically altering CSF proteome via active secretion of cytokines or metallopeptidases [55,83,84]. The CP has been also suggested as the primary site for the initiation of CNS inflammation, allowing free passage of immunocompetent cells from the blood into the CP stroma and their ensuing infiltration of the CSF [77,85,86]. This process is mediated by the upregulation of locally-secreted factors forming gradients, homing immune cells towards the CP epithelium, which exhibits disrupted organization allowing their paracellular passage into the CSF [87,88,89]. Interestingly, upon inflammation, leukocytes present in the CSF can invade the CP, suggesting the possibility of two-way trafficking of immunocompetent cells across the CP epithelium [90]. Considering the scope of effects associated with the CP-mediated secretion of immunomodulators and its role in the facilitation of leukocyte entry into the brain [91], the CP has been established as a central regulator of neuroinflammatory processes within the brain, raising important questions regarding the immune privilege of the CNS.

By virtue of its strategic location at the centre of the brain ventricular system, possession of barrier-like properties enabling tight control over CSF content, close contact with blood-borne signals, robust secretory capacity, and ability to sense changes in the local environment, the CP is uniquely poised to act as a master regulator of long-range signalling in the CNS. The CP thus acts as a principal nexus for integration and transmission of signals along the brain–body axis.

## 4. The CP–CSF Signalling Axis and Its Key Mediators

Deciphering the identity of molecular components underlying the diversity of CSF-mediated signalling along with the identification of the key sources of these factors has become undoubtedly one of the most exciting directions pursued in the field of the CSF research [7,92], with important implications for improvement of current therapies focused on brain regeneration [93].

Emerging knowledge of CP transcriptome and proteome profiles together with growing insight into the CSF content brought into the spotlight the CP as an important source for a plethora of biologically-active compounds found in the CSF at different developmental stages and physiological states [6,27,60]. Elucidation of the molecular identity of these substances is crucial for better understanding of the numerous ways through which CSF modulates key biological processes such as proliferation and neuronal viability during both embryogenesis and adulthood [94,95]. The different types of signalling molecules that were shown to be secreted by the CP to the CSF are summarized in Table 1.

While the roles for most of these compounds remains unknown, some initial findings indicate their involvement in diverse biological processes. We have previously shown that Wnt-5a, secreted by embryonic 4V CP, is able to influence the morphogenesis of the developing cerebellum [64]. Likewise, bone morphogenetic protein 5 (BMP-5), a member of the BMP family of signalling proteins, secreted by the adult LV CP was recently shown to function as a potent activator of adult neurogenesis [6]. By secretion of morphogen antagonists such as Wnt antagonist sFRP-1, CP secretome can further shape and fine-tune the signalling gradient of the growth factors contained in the CSF [109]. In addition to signalling peptides, the CP functions as source of other types of bioactive molecules, such as transcription factor Homeobox protein OTX2 (OTX2), which has been implicated in the regulation of neuroblast migration and integration of new-born neurons in the olfactory bulb [115]. Remarkably, blocking of the extracellular OTX2 was able to disrupt maturation of parvalbumin inhibitory neurons and expression of plasticity genes in the visual cortex of the adult mouse [116].

Further expanding the array of secreted substances, the CP has been recently established as the major source of another important class of regulatory molecules contained in the CSF - microRNAs [117]. Given its role in the modulation of expression of genes underlying cell cycle progression and differentiation, specific inhibition of the action of microRNA-204 (miR-204) derived from CP was linked to the depletion of adult quiescent neural stem cells (qNSC). Another microRNA, miR-128 [118], highly expressed by the CP, has been previously associated with various aspects of differentiation of adult neuronal progenitors [119]. In addition to the CPe, single-cell analysis of the embryonic and adult CP has demonstrated expression of genes encoding secreted factors, in additional cell types residing in the CP stroma such as fibroblasts expressing *Rbp4* or *Wisp1*, or endothelial cells producing NT3, raising an interesting possibility that these cell subpopulation actively contribute to the spectrum of signalling molecules released from the CP into CSF [49,107].

The CP is also involved in active transport of various blood-borne molecules, including cerebral transport of various micronutrients such as folate or iron, both essential for proper brain development [113,120]. In another example, megalin, a multiligand binding protein, secreted by the CPe has been shown to participate in the transport of IGF1 across the BCSFB and its release into the CSF [103,121].

Further exploration of the ability of the CP to shape CSF content led to recent findings that have established the CP as the key producer of binding proteins and multimolecular complexes regulating the extracellular transport of various signalling molecules via CSF [122,123]. The wide spectrum and chemical diversity of biologically active molecules (proteins, peptides, small molecules, and nucleic acids) produced by the CP opened the interesting question about what their cargo in the CSF is. Exosomes and lipoprotein complexes emerged as the most relevant candidate transport mechanisms.

### 4.1. Exosomes in CP–CSF Signalling

Exosomes represent a novel class of membranous extracellular vesicles increasingly recognized for their role as messengers involved in the long-range distribution of various compounds with regulatory functions [124], thus being able to affect a wide array of physiological and pathological processes in various tissues including the brain [125]. Proteomic analysis of CSF clearly shows enrichment of CSF exosomes for CP-specific proteins [126]. Evidence for active release of exosomes by the CP into the CSF came from study of the effects of systemic inflammation. Interestingly, CP-derived exosomes contained various miRNAs such miR-146a and miR-155, expressed by the CPe, which were able to cross the ependymal layer and be taken up by astrocytes and microglia in the brain parenchyma [105]. Increased levels of different miRNAs associated with CSF exosomes were also recently identified as biomarkers for various neuropathological conditions such as Parkinson’s disease or epilepsy [127,128], indicating that examination of the active secretion of different miRNAs from the CP may provide further insights of the underlying pathophysiology. Moreover, selective inhibition of miRNA expression using novel CP-targeting approaches opens interesting possibilities for further improvements in the treatment of numerous CNS-related disorders [38].

Interestingly, exosomes’ cargo varies over time, a process that may contribute to the age-dependent changes in the CNS function [129]. Exosomes were also suggested as a possible extracellular carrier for various signalling factors, as described for SHH released from embryonic 4V CP into the CSF [130]. Furthermore, exosomes generated by the CPe were also demonstrated to provide a transport mechanism for distribution of nutrients such as folate within the brain [120]. Moreover, exosomes secreted by the adult CPe can be hijacked to serve as a vehicle for the spread of virus infection from periphery to the CNS as shown for human polyomavirus (JCPyV) [131]. Intriguingly, CP-mediated release of exosomes may underlie transmission of SARS-coronavirus 2 (SARS-CoV-2) within the CNS as the CP displays relatively-high expression levels of ACE2 receptor, which is engaged by SARS-CoV-2 for active invasion of host cells [132,133,134,135].

### 4.2. Lipoprotein Complexes in CP–CSF Signalling

Lipoproteins represent another important group of extracellular particles produced by the CP [136]. Lipoproteins consist of a lipidic core surrounded by an outer layer that consists of hydrophobic lipids and apolipoproteins, which establish a special class of proteins with scaffolding function [137]. Lately, lipoproteins have been appreciated as being more than mere vehicles for transport of lipids, as evidenced by their capacity to carry various bioactive molecules with a central role in brain biology [138,139]. First, the CPe is the site for production of two of the most abundant apolipoproteins present in the CSF, namely apolipoprotein E (ApoE) and ApoJ [96,140,141,142]. Moreover, the CP was recently identified as the main entry point for lipoproteins containing ApoA-1 from the blood stream into the CNS [10]. Further, central components of molecular machinery underlying lipoprotein biogenesis, ABCA1 and ABCG1 [143], are expressed in the CP both in the embryonic development and in the adulthood [144,145]. Interestingly, apolipoprotein distribution varies between species with increased complexity observed in mammalian eCSF compared to avian eCSF, which has been hypothesized to reflect the more intricate neural architecture and synaptic plasticity seen in mammals [146]. Direct comparison of lipoproteins between CSF and blood revealed distinct pattern of posttranslational modifications, that might reflect the distinct functional properties and tissue-specific roles played by lipoproteins [147].

Underscoring the signalling potential of lipoproteins, low-density lipoproteins isolated from eCSF are responsible, to a significant extent, for neurogenic activity of the eCSF [148]. On the other hand, changed levels of high-density lipoproteins in CSF were linked to the pathophysiology of various neurodegenerative diseases [149]. In the recent years, several morphogens produced by the CPe were identified to associate directly with lipoproteins, including SHH [63,150], FGFs [98,151], or Wnts [62]. Moreover, association with lipoproteins, impaired in a mutated version of TREM2 receptor, a risk factor for AD, leads to suppression of lipoprotein-bound β-amyloid (Aβ) uptake in the CP [152,153].

Similarly to exosomes, lipoproteins were also shown to incorporate and actively transport miRNAs [154]. Furthermore, CSF lipoproteins are able to harbour proteins that can serve as interacting partners for morphogens, such as the Wnt ligand-binding partner, afamin. [155,156]. Heparan sulfate proteoglycans (HSPGs) represent a class of membrane-bound receptors for various ligands with the ability to shape the growth factor gradient [157], which can be also actively released into extracellular space including CSF [158]. Interestingly, it has been shown that the HSPG protein, glypican, found in CSF can bind directly to lipoprotein particles and contribute to adult neurogenesis [159,160]. In addition SHH, a morphogen secreted by the embryonic CP [63], was shown to associate with glypican-bound lipoprotein particles with important functions related to its internalization and signalling [160].

Aside from serving solely as scaffolding proteins for lipoproteins, apolipoproteins recently emerged as regulatory molecules in their own right with importance for the brain function [161]. Interaction of APOE with low-density lipoprotein receptor-related protein 1 (LRP1) was shown to affect differentiation of cortical and spinal cord neural stem cells progenitors [162]. Competition between APOE and tau protein for LRP1 receptor binding, which is involved in AD pathophysiology, has been recently demonstrated as a mechanism for reduction of tau uptake and its subsequent spread within the CNS [163]. ApoJ has been recently proposed as a molecular compound protecting against Aβ-mediated induction of Ca^2+^ influx into neuronal cells probably as a function of the ability of ApoJ to directly bind to Aβ in the CSF [164,165]. Moreover ApoJ was revealed to directly interact with Wnt-5a protein released by the CP into the CSF during embryogenesis [62]. On the other hand, decreased CSF levels of ApoA1 were revealed as biomarkers associated with increased risk of neurodegenerative diseases such as AD [166]. Moreover, intravenous injection of recombinant ApoA1 was shown to efficiently reduce Aβ load in the AD mouse model [167], thus identifying manipulation of CSF apolipoproteins as a promising strategy for future therapeutic applications for brain-related diseases. Thus, apolipoproteins produced by the CP and released into the CSF, tethered to lipoproteins, are linked to various aspects of CNS development, homeostasis, and pathology of neurodegenerative diseases. Nevertheless, it should be noted that a significant portion of the neurogenic effects associated with lipoprotein particle-mediated signalling can be attributed to their role as vehicles for the distribution of various lipid species and maintenance of lipid homeostasis in the brain [136].

## 5. The Target Brain Regions of CP–CSF Signalling

CSF-mediated signalling and neurogenesis are intimately linked processes with fundamental role in the development and homeostasis of the CNS. This is highlighted by the fact that neural precursors remain in close contact with CSF throughout life [168]. Neuroepithelium consisting of neuronal progenitors, which segregate to the ventricular zone upon neural tube closure and concomitant ventricular system formation, represent the chief source of cells that will give rise to the entire CNS [169]. Ample evidence collected over the last two decades has clearly established CSF as a crucial signalling component underpinning the key aspects of neuroepithelial behaviour [5,17,170]. Signalling factors such as Semaphorin-3B, released by the embryonic CP were shown to affect orientation of the mitotic spindle and apicobasal polarity, thus controlling division of neural progenitors [23]. Neuronal progenitors are characterized by the presence of sensory primary cilia enabling cells to sense and respond to the instructive cues present in CSF [2]. Several growth factors, including SHH and IGF-I, previously shown to be actively secreted by the CP into CSF [63,121], signal via primary cilia providing a possible mechanism, whereby CSF may directly regulate the fate of neuroepithelial cells [171,172]. Apart from simply regulating differentiation of neuronal progenitors, CSF-transported signalling factors were also shown to function as important local morphogenic regulators driving acquisition of regional identity during brain development [26,173]. Furthermore, molecular heterogeneity linked to CP positional identity, may result in generation of locally-restricted gradients of signalling molecules [62,63], thus contributing to the patterning effect of CSF observed during brain development.

Adult qNSCs, reside in two main anatomically-restricted germinal regions, the subventricular zone (SVZ), localized in the wall of the lateral ventricles, and the SGZ in the hippocampus [174,175]. qNSCs residing in the SVZ display multiple features facilitating their capacity to sense and be directly regulated by signals present in the CSF. One of the key morphological hallmarks of qNSCs is the presence of a short apical extension, allowing direct contact with the CSF. Akin to embryonic neuronal progenitors, the apical endings of qNSCs contain a primary cilium allowing these cells to actively sense the composition of the CSF [176]. Consistent with their shared embryonic origin, both neuronal precursors and qNSCs exhibit high levels of vascular cell adhesion molecule-1 (VCAM1) that plays a vital role in qNSC fate determination during embryogenesis and their maintenance in adulthood [101,177]. Moreover, VCAM1 has been shown to play the role of an environmental sensor as it can be upregulated in response to increased CSF levels of interleukin 1β and SDF1, which are highly expressed and actively secreted by adjacent LV CP [6,101]. In a similar fashion to that observed during embryonic brain patterning [178], behaviour of the adult stem cell niche can also be modulated by local gradients generated by CP-derived signalling molecules, such as chemorepulsive factor Slit2, which acts as a guidance molecule underlying long-distance migration of neuroblasts from the SVZ to the olfactory bulb [110]. Adding a further layer of complexity, recent advances using single cell analysis revealed not only spatial- but also gender-specific transcriptomic signatures in the SVZ [179]. Interestingly, a regionalized and sex-dependent pattern of expression was also associated with genes encoding various receptors and secreted molecules [179]. Notum, a secreted extracellular suppressor of Wnt ligands, displays SVZ-subdomain-specific expression [180], raising an interesting possibility of spatially-restricted regulation of Wnt pathway activation via localized deactivation of Wnt ligands, which were previously shown to be secreted from the CP into CSF [62,111]. Corroborating these findings, transthyretin (TTR), a thyroid hormone transporter predominantly produced by the CP [112], exhibits a gender-specific role in the control of neurogenesis in the SVZ that is restricted to a specific subdomain of the SVZ [181], which is in line with the gender-based differences in CP-derived TTR levels detected in the CSF [71,182]. Molecular heterogeneity linked to the receptor repertoire can be also observed between SVZ and SGZ, indicating intrinsic differences in the sensitivity of adult germinal centres to signalling factors presented via CSF [183,184]. This is supported by recent finding showing expression of LRP2, receptor for various morphogens, including BMPs and Wnts that were previously detected in CSF, to be restricted only to SVZ and devoid from SGZ. This was further confirmed by SVZ-specific effects of LRP-2 ablation on the regulation of neurogenesis [185,186]. Importantly, it has to be noted that additional cell types residing in the SVZ, but lacking direct contact with CSF, can also be a target of the CSF-borne signalling factors and complexes [105]. All these lines of evidence highlight the importance of CSF as a source of signalling factors in the CNS, which is preserved across the whole lifespan of an organism.

## 6. Concluding Remarks

Direct contact between neurogenic brain regions and CSF is the hallmark of CNS biology throughout life. CSF is the chief source of trophic factors and instructive cues underlying the key aspects of embryonic development and CNS patterning. The importance of CSF is also conserved in adulthood, when it plays a key role in the regulation of adult neurogenesis and the perturbation of its content is involved in numerous pathological conditions. Considering major breakthroughs in our understanding of CP function, it is becoming increasingly evident that the CP is the major player regulating signalling properties of CSF. This view is further emphasized by a growing list of signalling factors and transporting vesicles either directly produced in the CP or actively transferred from blood across the CP, which acts as a selective barrier between blood circulation and CSF (summarized in Figure 1). Importantly, these factors and vesicles have been linked to a myriad of aspects of brain biology during development and in adulthood. As a result of age progression, active infections or changes of physiological states, CP transcriptome and secretome can undergo dramatic changes, thus highlighting the CP as a vital component involved in the modulation of crucial biological processes. Considering the overarching influence of the CP as the signalling hub of the brain, the recent emergence of experimental approaches for closer examination and manipulation of various facets of CP secretory activity promises to shed light on various outstanding challenges facing the field. Virus-based vectors, have been described as an exciting new tool for targeted and highly efficient gene delivery, enabling gene manipulation in the CP [187] and providing a powerful technique for modulation of CNS biological functions via specific alterations of CP proteome [188]. In addition, there is growing scientific interest in leveraging the potential of exosomes and lipoproteins for brain-targeted drug delivery [189,190]. At the same time, organoids have been recently recognized as an interesting model to study development of the CP and CSF production [7,191], as evidenced by presence of functional CP-like structures connected to fluid-filled cavities mimicking the functional CP–CSF interface [192,193]. Given the possibility of genetic manipulation, organoids represent a tractable model for the investigation of the molecular mechanism underlying various pathologies associated with impaired CP secretion and CSF production [130,194,195]. Thus, examination of CP secretory properties using a wide array of newly-developed molecular techniques represents an alluring avenue for future research with important implications for our understanding of brain biology across life and improvement of medical interventions aimed at the underlying causes of various developmental or neurodegenerative conditions.

## Figures and Tables

**Figure 1 ijms-21-04760-f001:**
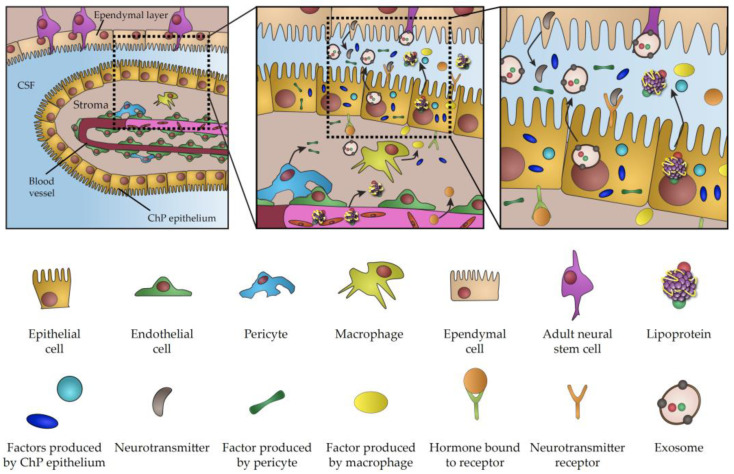
Schematic representation of choroid plexus secretome.

**Table 1 ijms-21-04760-t001:** List of signalling factors secreted by the choroid plexus (CP).

Molecule	Category	Biological Function	Species	Stage	References
ApoE	Apolipoprotein	Lipid transport, Alzheimer’s disease (AD) protection	Mouse	Adult	[96]
ApoJ	Apolipoprotein	Lipid transport, AD protection	Mouse	Embryonic	[62]
sAPP	Secreted protein	Adult neurogenesis	Mouse	Adult	[38]
Augurin	Hormone	Cell proliferation	Mouse	Adult	[73]
BMP-5	Growth factor	Adult neurogenesis	Mouse	Adult	[6]
CT-1	Growth factor	Gliogenesis regulation	Rat	Adult	[97]
FGF2	Growth factor	CP embryogenesis	Human, mouse	Embryonic	[98]
Hepicidin	Transporter protein	Brain iron homeostasis	Mouse, rat	Adult	[99]
IGF-II	Growth factor	Embryonic neurogenesis	Mouse	Embryonic	[5]
IGFBP-2	Secreted protein	IGF signalling regulator	Rat	Adult	[100]
IL-1 beta	Cytokine	Adult neurogenesis	Mouse	Adult	[101]
αKlotho	Secreted Enzyme	Anti-aging effects	Human, rat	Adult	[102]
Megalin	Heparan sulfate proteoglycan (HSPG)	Ligand transport	Human	Adult	[103]
Melatonin	Hormone	Sleep–wake cycle regulation	Rat	Adult	[104]
miR-146a	microRNA	Inflammatory response	Mouse	Adult	[105]
mIR-204	microRNA	Adult neurogenesis	Mouse	Adult	[106]
NT-3	Growth factor	Adult neurogenesis	Mouse	Adult	[107]
Homeobox protein OTX2	Transcription factor	Adult neurogenesis	Mouse	Adult	[108]
Semaphorin-3B	Secreted protein	Neuroepithelium proliferation	Mouse	Embryonic	[23]
sFRP-1	Secreted protein	AD pathogenesis	Human	Embryonic	[109]
SHH	Growth factor	Cerebellum development	Mouse	Embryonic	[63]
Slit-1	Secreted protein	Adult neurogenesis	Mouse	Adult	[110]
Tgm2	Secreted enzyme	Embryonic development	Mouse	Embryonic	[111]
Transthyretin	Transport protein	Adult neurogenesis	Rat	Adult	[112]
Transferrin	Transporter protein	Brain iron homeostasis	Rat	Adult	[113]
VEGF	Growth factor	Angiogenesis	Canine	Adult	[114]
Wnt-5a	Growth factor	Cerebellum morphogenesis	Mouse	Embryonic	[62]

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
