# Peer review of "Choroid Plexus: The Orchestrator of Long-Range Signalling Within the CNS"

_ijms, 2020, doi:10.3390/ijms21134760_

Round 1

Reviewer 1 Report

I congratulate the Authors for drafting this review article which will be of benefit to the IJMS audience. The effect of CSF and CP in the regulation and propagation of the CNS inflammatory cascade is the only underrepresented aspect of this manuscript. With the emerging data of the glymphatic system and the free movement of T/B cells through the lymph, CP, and the brain, the idea of the CNS as immune-privileged tissue is now being questioned. In terms of completeness, I would only suggest adding a short section on the role of CSF and CP in the CNS immune control/tolerance. 

The manuscript is well written and describes the role of CSF and CP during both the developmental and adult stages of CNS function. The role of CP in the processes of CNS immunosurveillance is an emerging topic in the field and thus, this review is timely. My comments for further improvement are shown hereafter:

  • Description of the literature search strategy utilized for this review is needed. Which sources and MeSH terms were used? How many articles were retrieved and cataloged?
  • In order to provide a complete preview of the topic, addition of the role of the CSF and CP in the regulation and propagation of the CNS inflammatory may be warranted. With the emerging data of the glymphatic system and the free movement of T/B cells through the lymph and CSF, the idea of the CNS as immune-privileged tissue is now being questioned. I would suggest adding a short section on the role of CSF and CP in the CNS immune control/tolerance. For example, the choroid plexus may be a potential site where the initiation of CNS inflammatory disease can occur (the literature describes the CP as place for Th17 cell migration and trafficking). Moreover, chronic inflammatory diseases can utilize the stroma of the choroid plexus as a site of continuous cytokine production site that can sustain the pathology despite the intact integrity of the BBB (compartmentalized activation). Also, CSF acts as a long-range transporter of inflammatory cytokines and cells causing inflammatory and neurodegenerative pathology to tissues neighboring the ventricles or CSF spaces. Due to the continuum provided by the CSF, potentially inflammatory and toxic molecules like cytokines and ceramides.
  • Adding sentences regarding the translational value of each experimental findings (miRs or Apos) would be beneficial. For example, commentary regarding APO-A1 mimetic therapy in neurodegenerative disease (data from ApoA-I Milano, ETC-216 from Pfizer which was renamed MDCO-216). Also, HDL and ApoA have been also linked to neuroinflammatory/neurodegenerative diseases with recent large human trials targeting that pathway (statin use in secondary-progressive MS).

Author Response

I congratulate the Authors for drafting this review article which will be of benefit to the IJMS audience. The effect of CSF and CP in the regulation and propagation of the CNS inflammatory cascade is the only underrepresented aspect of this manuscript. With the emerging data of the glymphatic system and the free movement of T/B cells through the lymph, CP, and the brain, the idea of the CNS as immune-privileged tissue is now being questioned. In terms of completeness, I would only suggest adding a short section on the role of CSF and CP in the CNS immune control/tolerance. 

Rely: We thank reviewer for the positive assessment of the review. Requested section, discussing the role of CP-CSF axis in the regulation of neuroinflammation, has been added to the text and can be now found as an additional paragraph on the page 5 in the re-submitted version of the review (refs. 81 – 94). We believe that this novel section clearly outlines the important role of CP secretome in the regulation of neuroinflammation within CNS and contributes to improved overall quality of the review in terms of providing a fuller picture of the CP function in the regulation of long-range signaling within brain.

The manuscript is well written and describes the role of CSF and CP during both the developmental and adult stages of CNS function. The role of CP in the processes of CNS immunosurveillance is an emerging topic in the field and thus, this review is timely. My comments for further improvement are shown hereafter:

Reply: We thank Reviewer 1 for the positive evaluation of our manuscript and the suggestions that we believe improved significantly the overall quality of the manuscript. In the following text we provide point-by-point reply to individual requests.

Description of the literature search strategy utilized for this review is needed. Which sources and MeSH terms were used? How many articles were retrieved and cataloged?

Reply: The relevant literature has been searched using PubMed and Google Scholar with. Individual studies were manually evaluated for relevance and novelty. Where possible MeSH heading were used to retrieve literature sources including: choroid plexus (Unique ID: D002831) , cerebrospinal fluid (Unique ID: D002555), exosomes (Unique ID: D055354), lipoproteins (Unique ID: D008074), neurogenesis (Unique ID: D055495), embryonic development (Unique ID: D047108), Alzheimer disease (Unique ID: D000544), microRNAs (Unique ID: D035683),

In order to provide a complete preview of the topic, addition of the role of the CSF and CP in the regulation and propagation of the CNS inflammatory may be warranted. With the emerging data of the glymphatic system and the free movement of T/B cells through the lymph and CSF, the idea of the CNS as immune-privileged tissue is now being questioned. I would suggest adding a short section on the role of CSF and CP in the CNS immune control/tolerance. For example, the choroid plexus may be a potential site where the initiation of CNS inflammatory disease can occur (the literature describes the CP as place for Th17 cell migration and trafficking). Moreover, chronic inflammatory diseases can utilize the stroma of the choroid plexus as a site of continuous cytokine production site that can sustain the pathology despite the intact integrity of the BBB (compartmentalized activation). Also, CSF acts as a long-range transporter of inflammatory cytokines and cells causing inflammatory and neurodegenerative pathology to tissues neighboring the ventricles or CSF spaces. Due to the continuum provided by the CSF, potentially inflammatory and toxic molecules like cytokines and ceramides.

Reply: We included an individual paragraph dedicated to the neuroinflammation (lines: 199 – 218; references: 80 – 93) in the context of Choroid plexus – cerebrospinal fluid signaling axis. We believe that in its current form this paragraph addresses sufficiently all the aspects of neuroinflammation highlighted by the reviewer.

Adding sentences regarding the translational value of each experimental findings (miRs or Apos) would be beneficial. For example, commentary regarding APO-A1 mimetic therapy in neurodegenerative disease (data from ApoA-I Milano, ETC-216 from Pfizer which was renamed MDCO-216). Also, HDL and ApoA have been also linked to neuroinflammatory/neurodegenerative diseases with recent large human trials targeting that pathway (statin use in secondary-progressive MS).

Reply: We added several sentences addressing the translational value of the mentioned findings as suggested by the reviewer. These include the portions of the text highlighting the translational importance of exosomal miRNAs (lines: 281 – 286; references: 38, 130, 131) and apolipoproteins (lines: 340 – 344; references: 170-171).

Reviewer 2 Report

The manuscript presented by Kaiser and Bryja is a very interesting review, in which authors extensively described the very important role that Choroid plexus tissue is playing in many aspects of signaling and developing of CNS; a functional view for this tissue not extensively described before. The common general way of seen Choroid plexus is as the organ uncharged of producing CSF, to fill the ventricle cavities, pial and central spinal cord spaces, and providing a protective role as a cushion effect avoiding mechanical contusions. However, the manuscript presented here described an immense variety of factors either directly produced by choroid plexus or produced in other places bring by the blood and then transported by choroid plexus to the CSF.

My general opinion is that the manuscript is very interesting, well organized and written, and put in value the real relevance of Choroid plexus as coordinator of many process in CNS.

My only concern is regarding the data in table 1. In which is not indicated Which specie (human, mice, etc) and age (embryonic, adult) of choroid plexus information is been posted there.

Author Response

My general opinion is that the manuscript is very interesting, well organized and written, and put in value the real relevance of Choroid plexus as coordinator of many process in CNS.

My only concern is regarding the data in table 1. In which is not indicated Which specie (human, mice, etc) and age (embryonic, adult) of choroid plexus information is been posted there.

Reply: We are grateful to the reviewer for practical suggestion and for the positive assessment of the review. We implemented requested modifications to the Table 1, which now contains information regarding both the developmental stage (Age) and used species for each study listed in the Table 1. We believe that this modification clearly improved the overall quality of the review.